# FORECASTING PROBABILITY DISTRIBUTION OF NON-LINEAR TIME SERIES

**Kyongmin Yeo, Igor Melnyk, Nam Nguyen & Eun Kyung Lee**
IBM T.J. Watson Research Center
Yorktown Heights, NY, USA
{kyeo,nnguyen,eunkyung.lee}@us.ibm.com, igor.melnyk@ibm.com

## ABSTRACT

We propose DE-RNN to learn the probability density function (PDF) of a nonlinear time series, and compute the temporal evolution of the PDF for a probabilistic forecast. A Recurrent Neural Network (RNN) based model is employed to learn a nonlinear operator for temporal evolution of the stochastic process. We use a softmax layer for a numerical discretization of a PDF, which transforms a function approximation problem to a classification problem. Explicit and implicit regularization strategies are introduced to impose a smoothness condition on the estimated probability distribution. A multiple-step forecast is achieved by computing the time evolution of PDF.

## 1 INTRODUCTION

We consider a problem of learning a PDF of a noisy nonlinear dynamical system, or a stochastic process with an underlying nonlinear structure. The stochastic process is given as

$$\hat{y}_t = y(t) + \epsilon_t, \tag{1}$$

in which $\epsilon_t$ is a noise process and $y(t)$ is an underlying nonlinear dynamics, e.g.,

$$\frac{\partial y}{\partial t} = \mathcal{F}(y(t), y(t - \tau), \boldsymbol{u}(t)). \tag{2}$$

Here, $y(t)$ is a continuous process, $\mathcal{F}$ is a nonlinear operator, $\tau$ is a delay-time parameter, and $\boldsymbol{u}(t)$ is an exogenous forcing, such as control parameters. In (2) and (1), $\mathcal{F}$ is assumed unknown, and we do not assume any distributional properties of $\epsilon_t$, but assume the knowledge of the control $\boldsymbol{u}(t)$. We are interested in computing temporal evolution of PDF of $\hat{y}$, given the observations up to time step $t$, i.e., $p(\hat{y}_{t+n}|\widehat{\boldsymbol{Y}}_{0:t}, \boldsymbol{U}_{0:t+n-1})$ for $n \geq 1$, where $\widehat{\boldsymbol{Y}}_{0:t} = (\hat{y}_0, \cdots, \hat{y}_t)$ is a trajectory of the past observations and $\boldsymbol{U}_{0:t+n-1} = (\boldsymbol{u}_0, \cdots, \boldsymbol{u}_{t+n-1})$ consists of the history of the known control actions, $\boldsymbol{U}_{0:t-1}$, and a future control scenario, $\boldsymbol{U}_{t:t+n-1}$. Hereafter, we use DE-RNN for the proposed RNN model, considering the similarity with the density estimation. Note that DE-RNN has a direct relevance to many applications in manufacturing processes (Lasi et al., 2014).

## 2 DE-RNN FOR NOISY DYNAMICAL SYSTEM

DE-RNN is based on the Long Short-Term Memory (LSTM) network (Hochreiter & Schmidhuber, 1997; Gers et al., 2000) for the modeling of time evolution. LSTM consists of a set of nonlinear transformations of the input variable $\boldsymbol{x}_t = (\hat{y}_t, \boldsymbol{u}_t)$ and the output of the previous time step, $\boldsymbol{h}_{t-1}$. A single layer LSTM model can be summarized by the following set of equations;

$$\boldsymbol{s}_t = \Psi_s(\boldsymbol{x}_t, \boldsymbol{s}_{t-1}, \boldsymbol{h}_{t-1}), \ \boldsymbol{h}_t = \Psi_h(\boldsymbol{x}_t, \boldsymbol{s}_t, \boldsymbol{h}_{t-1}), \ \boldsymbol{P}_{t+1} = \Psi_p(\boldsymbol{h}_t), \tag{3}$$

in which $\boldsymbol{s}_t$ and $\boldsymbol{h}_t$ are the internal state and output of LSTM, respectively.

### 2.1 DISCRETIZATION OF PROBABILITY DENSITY FUNCTION

We first consider the problem of modeling the PDF of a random variable $\hat{y}$, given an input $x$, i.e., $p(\hat{y}|x)$. The obtained results can be directly applied to the original problem, $p(\hat{y}_{t+1}|\widehat{\boldsymbol{Y}}_{0:t}, \boldsymbol{U}_{0:t})$.

Let $\boldsymbol{\alpha} = (\alpha_0, \cdots, \alpha_K)$ denote a set of real numbers, such that $\alpha_{i-1} < \alpha_i$ for $i = 1, \cdots, K$, which defines $K$ disjoint intervals, $\mathcal{I}_i = (\alpha_{i-1}, \alpha_i)$. Then, a discrete probability can be defined

$$p(k|x) = \int_{\mathcal{I}_k} p(\hat{y}|x)dy, \text{ for } k = 1, \ldots, K. \tag{4}$$

It is clear that $p(k|x)$ is a numerical discretization of the continuous PDF, $p(\hat{y}|x)$. The discrete probability $p(k|x)$ can be modeled by a softmax layer ($\boldsymbol{P}$) as an output of $\Psi_p$ in (3) such that

$$p(k|x) = P_k, \text{ for } k = 1, \ldots, K. \tag{5}$$

The discretization naturally leads to the conventional cross-entropy (CE) minimization. Suppose we have a data set, $\boldsymbol{D}_R = \{(\hat{y}_i, x_i); \hat{y}_i \in \mathbb{R}, x_i \in \mathbb{R}, \text{ and } i = 1, \ldots, N\}$. We can define a mapping $\mathcal{C} : \mathbb{R} \to \mathbb{N}_+$ such that $\mathcal{C}(\hat{y}) = k$, if $y \in \mathcal{I}_k$. Then, $\boldsymbol{D}_R$ can be easily converted to a new data set for target labels, $\boldsymbol{D}_C = \{(c_i, \hat{y}_i, x_i); c_i \in \mathbb{N}_+, \hat{y}_i \in \mathbb{R}, x_i \in \mathbb{R}, \text{ and } i = 1, \ldots, N\}$, where $c_i = \mathcal{C}(\hat{y}_i)$. $\boldsymbol{D}_C$ provides a training data set for the following CE loss,

$$CE = -\sum_{n=2}^{N}\sum_{k=1}^{K} \delta_{c_n k} \log P_{n,k} = -\sum_{n=2}^{N} \log P_{n,c_n}, \tag{6}$$

in which $\boldsymbol{P}_n = (\Psi_p \circ (\Psi_h \circ \Psi_s))(\boldsymbol{x}_{n-1}, \boldsymbol{s}_{n-2}, \boldsymbol{h}_{n-2})$. Note, however, that the CE minimization does not explicitly guarantee the smoothness of the estimated distribution. To address this issue, we propose a regularized CE loss.

### 2.1.1 Regularization of Cross-Entropy Loss

To explicitly impose the smoothness between the classes, we propose to use a regularized cross-entropy (RCE) minimization, defined by the following loss function

$$\text{RCE} = \sum_{n=2}^{N}\left\{\sum_{k=1}^{K} -\delta_{c_n k}\log P_{n,k} + \lambda\left(\boldsymbol{L}\boldsymbol{P}_n\right)^T \boldsymbol{L}\boldsymbol{P}_n\right\}, \boldsymbol{L} = \begin{bmatrix} 1 & -2 & 1 & 0 & \cdots & 0 \\ 0 & 1 & -2 & 1 & \cdots & 0 \\ \dotfill \\ 0 & \cdots & 0 & 1 & -2 & 1 \end{bmatrix}. \tag{7}$$

where $\lambda$ is a penalty parameter and the Laplacian matrix $\boldsymbol{L} \in \mathbb{R}^{K-2,K}$. RCE is similar to the penalized maximum likelihood solution for density estimation (Silverman, 1986), because

$$\boldsymbol{P}_n^T \boldsymbol{L}^T \boldsymbol{L} \boldsymbol{P}_n \sim \int \left[p''(\hat{y}|x)\right]^2 dy. \tag{8}$$

Alternative to adding an explicit regularization to CE, the smoothness can be achieved by enforcing a spatial correlation in the network output by using a convolution layer. Let $\widetilde{\boldsymbol{o}} \in \mathbb{R}^K$ denote the last layer of DE-RNN, which was the input to the softmax layer. We can add a convolution layer, $\boldsymbol{o} \in \mathbb{R}^K$, on top of $\widetilde{\boldsymbol{o}}$, such that

$$o_i = \sum_{j=1}^{K} \frac{1}{h} \exp\left[-\frac{1}{2}\left(\frac{i-j}{h}\right)^2\right]\widetilde{o}_j, \text{ for } i = 1, \cdots, K, \tag{9}$$

where the parameter $h$ determines the smoothness of the estimated distribution. Then, $\boldsymbol{o}$ is supplied to the softmax layer. Using (9), DE-RNN can now be trained by the standard CE. The implicit regularization, here we call convolution CE (CCE), is analogous to a kernel density estimation.

### 2.2 Multivariate Time Series

We propose to train a set of DE-RNNs to compute the joint PDF of a $l$-dimensional multivariate time series; $\hat{\boldsymbol{y}}_t = (\hat{y}_t^{(1)}, \cdots, \hat{y}_t^{(l)})$ by using the product rule,

$$p(\hat{\boldsymbol{y}}_{t+1}|\widehat{\boldsymbol{Y}}_{0:t}, \boldsymbol{U}_{0:t}) = p(\hat{y}_{t+1}^{(1)})\prod_{j=2}^{l} p(\hat{y}_{t+1}^{(j)}\big|\hat{y}_{t+1}^{(j-1)}, \cdots, \hat{y}_{t+1}^{(1)}, \widehat{\boldsymbol{Y}}_{0:t}, \boldsymbol{U}_{0:t}).$$

Note that, although it requires training $l$ DE-RNNs to compute the full joint PDF, there is no dependency between the DE-RNNs in the training phase. So, the set of DE-RNNs can be trained in parallel, which can greatly reduce the training time.

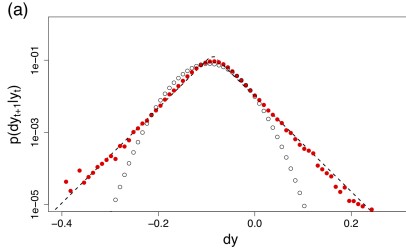 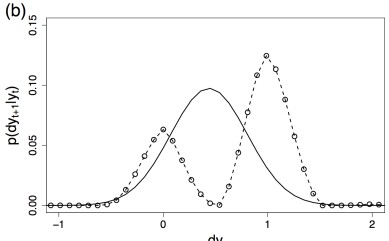

Figure 1: Comparison of DE-RNN and DeepAR(Flunkert et al., 2017) for (a) Mackey-Glass time series with Laplace noise and (b) CPU temperature data. In (a), the ground truth is shown in the dashed line. DE-RNN (●) and DeepAR (○) are denoted by circles. In (b), the circles denote DE-RNN, while the solid line is from DeepAR.

Table 1: Normalized errors of the Lorenz time series. DE-RNN results are compared with DeepAR (DAR), Gaussian Process (GP), and Vector AutoRegressive model (VAR.)

|            | DE-RNN | DAR   | GP    | VAR   |
|------------|--------|-------|-------|-------|
| $e_\mu$    | 0.134  | 0.140 | 0.506 | 0.917 |
| $e_\Sigma$ | 0.040  | 0.560 | 0.596 | 0.558 |

### 2.3 COMPUTING TIME EVOLUTION OF PROBABILITY DISTRIBUTION

Note that even though $\boldsymbol{H}_{t+1} = (\boldsymbol{s}_{t+1}, \boldsymbol{h}_{t+1})$ is computed from deterministic functions from data,

$$\boldsymbol{s}_{t+1} = \Psi_s(\hat{y}_{t+1}, \boldsymbol{H}_t), \ \boldsymbol{h}_{t+1} = \Psi_h(\hat{y}_{t+1}, \boldsymbol{s}_{t+1}, \boldsymbol{h}_t),$$

$\boldsymbol{H}_{t+1}$ is a random variable, because $\hat{y}_{t+1}$ is a random variable. A multiple-step forecast can be computed by repeatedly computing the time evolution of $\boldsymbol{H}_t$ as

$$p(\hat{y}_{t+n}|\widehat{\boldsymbol{Y}}_{0:t}, \boldsymbol{U}_{0:t+n-1}) = \int p(\hat{y}_{t+n}|\boldsymbol{h}_{t+n-1}) \prod_{i=t+1}^{t+n-1} p(\boldsymbol{H}_i|\boldsymbol{H}_{i-1}, \boldsymbol{u}_i) \, d\boldsymbol{H}_i. \tag{10}$$

The high dimensional integration in (10) can be evaluated by a Sequential Monte Carlo method.

## 3 EXPERIMENTS

DE-RNN is tested against three synthetic and two real data sets. For the synthetic data, a modified Cox-Ingersoll-Ross process, which is a multiplicative noise process, Mackey-Glass with non-Gaussian noise, and (multivariate) Lorenz times series with a Gaussian noise are used. For the real data, Mauna Loa $CO_2$ observations and CPU temperature of IBM Power System S822LC are used.

Figure 1 shows the probability distribution, $p(\hat{y}_{t+1}|\widehat{\boldsymbol{Y}}_{0:t})$, estimated by DE-RNN. It is shown that DE-RNN represents the Laplace distribution without any special modeling (Figure 1 a). The temperature in the CPU data set is discrete, because the resolution of the temperature sensor is $1°C$. Figure 1 (b) shows that DE-RNN well captures the bimodal distribution due to the $1°C$ sensor resolution.

Table 1 shows the normalized root mean-square errors in the expectation ($e_\mu$) and covariance ($e_\Sigma$) for the Lorenz time series. $e_\Sigma$ is defined by the Frobenius norm. It is shown that DE-RNN makes a very good prediction of both the expectation and covariance. The error in the covariance in DE-RNN is only about 4%. Because DeepAR and GP do not consider the off-diagonal components of the covariance matrix, $e_\Sigma$ of those models are much larger than DE-RNN.

Multiple-step forecasts of DE-RNN show that the prediction uncertainty by DE-RNN does not grow monotonically in time. For 1,500-step-ahead prediction of the CPU temperature, RMSE is only $0.83°C$, compared to $0.56°C$ of the one-step-ahead prediction.

The evaluation of DE-RNN on the synthetic and real data sets shows advantage of DE-RNN over the compared baselines.

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
