# OpenReview forum: "Forecasting probability distribution of nonlinear time series"
_ICLR.cc/2018/Workshop — Reject_

### Official Review · AnonReviewer3 · 2018-03-09
**Well-known combination of LSTM and discretisation of time series**

**Rating:** 5
**Confidence:** 4

**Review:**

This paper studies the use of LSTMs to do nonlinear time-series forecasting while explicitly expressing prediction uncertainty, by modeling the univariate or multivariate scalar values as probability distribution functions using multinomials over discretised time-series.

Following an LSTM by a softmax and replacing regression by a classification task is a well known trick for LSTM (please see the PixelRNN and derived papers, e.g., "Pixel recurrent neural networks" by Van Den Oord et al, 2016). As such, there is nothing novel in this paper.

What would have been interesting here would have been an exhaustive study of time series generation (iterated prediction) while sampling y^{hat}_{t+j} (for j > 1), a subject that is barely mentioned in section 2.3 when talking about Sequential Monte-Carlo. However, the authors get lost in long exposition in section 2.1 and 2.2. Is it necessary to introduce psi_s and psi_h? Does y^{hat} get used as input? Could equations (4) and (5) get replaced by saying: "we discretise the time series into categorical values"? The cross-entropy regularisation bit is novel, though.

The results seem good but miss an important baseline: plain LSTMs!

Figure 1 should use the same symbols for (a) and (b).

---

### Official Review · AnonReviewer2 · 2018-03-11
**Interesting work which could be potentially improved**

**Rating:** 5
**Confidence:** 3

**Review:**

This paper seeks to solve time-series PDF estimation problem. It applies a RNN to capture the temporal dependency over the past observations. It also uses a grid discretization to transform a PDF estimation formulation into a classification problem. Experiments show that the proposed approach perform favorably compared to DeepAR, AR and GP. However, the paper can be potentially improved in the following aspects. First, the scalability of DE-RNN is not discussed in the experiments. One can imagine that the number of grids determines the prediction performance, and due to the curse of dimensionality it will requires large number of training examples. The experiments only give results in a one-dimensional setting, which is not sufficiently convincing. Second, this paper seems to extend to multi-dimensional cases, which will make the training even harder. Moreover, the critical part of multi-variate time-series is that the interdependency among different dimensions is important to the prediction of each individual dimension. However, the paper totally ignores this dependency, which makes this setting not very useful in practice. Third, the paper proposes two ways of regularization but fails give the comparison of the two. It also lacks of training setting specification for the other baselines.

---

### Official Review · AnonReviewer1 · 2018-03-11

**Rating:** 6
**Confidence:** 3

**Review:**

This paper proposes an approach to non-linear time-series prediction that outperforms previous approach on several datasets.

---

### Decision · Program_Chairs · 2018-03-20
**ICLR 2018 Workshop Acceptance Decision**

**Decision:**

Reject

**Comment:**

Based on the reviews, this paper has not been accepted for presentation at the ICLR workshop. However, the conversation and updates can continue to appear here on OpenReview.